# ONLINE AND STOCHASTIC OPTIMIZATION BEYOND LIPSCHITZ CONTINUITY: A RIEMANNIAN APPROACH

**Kimon Antonakopoulos**
Inria, Univ. Grenoble Alpes, CNRS, Grenoble INP, LIG
38000 Grenoble, France
`kimon.antonakopoulos@inria.fr`

**E. Veronica Belmega**
ETIS UMR8051, CY University, ENSEA, CNRS, F-95000, Cergy, France
`belmega@ensea.fr`

**Panayotis Mertikopoulos**
Inria, Univ. Grenoble Alpes, CNRS, Grenoble INP, LIG
38000 Grenoble, France
`panayotis.mertikopoulos@imag.fr`

## ABSTRACT

Motivated by applications to machine learning and imaging science, we study a class of online and stochastic optimization problems with loss functions that are not Lipschitz continuous; in particular, the loss functions encountered by the optimizer could exhibit gradient singularities or be singular themselves. Drawing on tools and techniques from Riemannian geometry, we examine a *Riemann–Lipschitz* (RL) continuity condition which is tailored to the singularity landscape of the problem's loss functions. In this way, we are able to tackle cases beyond the Lipschitz framework provided by a global norm, and we derive optimal regret bounds and last iterate convergence results through the use of regularized learning methods (such as online mirror descent). These results are subsequently validated in a class of stochastic Poisson inverse problems that arise in imaging science.

## 1 INTRODUCTION

The surge of recent breakthroughs in machine learning and artificial intelligence has reaffirmed the prominence of first-order methods in solving large-scale optimization problems. One of the main reasons for this is that the computation of higher-order derivatives of functions with thousands – if not millions – of variables quickly becomes prohibitive; another is that gradient calculations are typically easier to distribute and parallelize, especially in large-scale problems. In view of this, first-order methods have met with prolific success in many diverse fields, from machine learning and signal processing to wireless communications, nuclear medicine, and many others [10, 34, 37].

This success is especially pronounced in the field of *online optimization*, i.e., when the optimizer faces a sequence of time-varying loss functions $f_t$, $t = 1, 2, \ldots$, one at a time – for instance, when drawing different sample points from a large training set [11, 35]. In this general framework, first-order methods have proven extremely flexible and robust, and the attained performance guarantees are well known to be optimal [1, 11, 35]. Specifically, if the optimizer faces a sequence of $G$-Lipschitz convex losses, the incurred min-max regret after $T$ rounds is $\Omega(GT^{1/2})$, and this bound can be achieved by inexpensive first-order methods – such as online mirror descent and its variants [11, 35, 36, 41].

Nevertheless, in many machine learning problems (support vector machines, Poisson inverse problems, quantum tomography, etc.), the loss landscape is *not* Lipschitz continuous, so the results mentioned above do not apply. Thus, a natural question that emerges is the following: *Is it possible to apply online optimization tools and techniques beyond the standard Lipschitz framework? And, if so, how?*

**Our approach and contributions.** Our point of departure is the observation that Lipschitz continuity is a property of *metric* spaces – not *normed* spaces. Indeed, in convex optimization, Lipschitz continuity is typically stated in terms of a global norm (e.g., the Euclidean norm), but such a norm is *de facto* independent of the point in space at which it is calculated. Because of this, the standard Lipschitz framework is oblivious to the finer aspects of the problem's loss landscape – and, in particular, any singularities that may arise at the boundary of the problem's feasible region. On the other hand, in general metric spaces, this is no longer the case: the distance between two points is no longer given by a global norm, so it is much more sensitive to the geometry of the feasible region. For this reason, if the (Riemannian) distance $\text{dist}(x, x')$ between two points $x$ and $x'$ becomes larger and larger as the points approach the boundary of the feasible region, a condition of the form $|f(x) - f(x')| = \mathcal{O}(\text{dist}(x', x))$ may still hold even if $f$ becomes singular at the boundary.

We leverage this observation by introducing the notion of *Riemann–Lipschitz* (RL) continuity, an extension of "vanilla" Lipschitz continuity to general spaces endowed with a Riemannian metric. We show that this metric can be chosen in a principled manner based solely on the singularity landscape of the problem's loss functions – i.e., their growth rate at infinity and/or the boundary of the feasible region. Subsequently, using a similar mechanism to choose a *Riemannian regularizer*, we provide an optimal $\mathcal{O}(T^{1/2})$ regret guarantee through the use of regularized learning methods – namely, "follow the regularized leader" (FTRL) and online mirror descent (OMD).

Our second contribution concerns an extension of this framework to stochastic programming. First, in the context of stochastic *convex* optimization, we show that an online-to-batch conversion yields an $\mathcal{O}(T^{-1/2})$ value convergence rate. Second, motivated by applications to nonconvex stochastic programming (where averaging is not a priori beneficial), we also establish the convergence of the method's last iterate in a class of nonconvex problems satisfying a weak secant inequality. Finally, we supplement our theoretical analysis with numerical experiments in Poisson inverse problems.

**Related work.** To the best of our knowledge, the first treatment of a similar question was undertaken by Bauschke et al. [3] who focused on deterministic, *offline* convex programs ($f_t = f$ for all $t$) without a *Lipschitz smoothness* assumption (i.e., Lipschitz continuity of the *gradient*, as opposed to Lipschitz continuity of the *objective*). To tackle this issue, Bauschke et al. [3] introduced a second-order "Lipschitz-like" condition of the form $\nabla^2 f \preccurlyeq \beta \nabla^2 h$ for some suitable Bregman function $h$, and they showed that Bregman proximal methods achieve an $\mathcal{O}(1/T)$ value convergence rate in offline convex problems with *perfect* gradient feedback.

Always in the context of deterministic optimization, Bolte et al. [8] extended the results of Bauschke et al. [3] to unconstrained non-convex problems and established trajectory convergence to critical points for functions satisfying the Kurdyka–Łojasiewicz (KL) inequality. In a slightly different vein, Lu et al. [25] considered functions that are also strongly convex relative to the Bregman function defining the Lipschitz-like condition for the gradients, and they showed that mirror descent achieves a geometric convergence rate in this context. Finally, in a very recent preprint, Hanzely et al. [17] examined the rate of convergence of an accelerated variant of mirror descent under the same Lipschitz-like smoothness assumption.

Importantly, all these works concern *offline*, deterministic optimization problems with *perfect* gradient feedback and regularity assumptions that cannot be exploited in an online optimization setting (such as the KL inequality). Beyond offline, deterministic optimization problems, Lu [24] established the ergodic convergence of mirror descent in stochastic non-adversarial convex problems under a "relative continuity" condition of the form $\|\nabla f(x)\| \le G \inf_{x'} \sqrt{2D(x', x)}/\|x' - x\|$ (with $D$ denoting the divergence of an underlying "reference" Bregman function $h$). More recently, Hanzely and Richtárik [16] examined the performance of stochastic mirror descent under a combination of relative strong convexity and relative smoothness / Lipschitz-like conditions, and established a series of convergence rate guarantees that mirror the corresponding rates for ordinary (Euclidean) stochastic gradient descent. Except for trivial cases, these conditions are not related to Riemann–Lipschitz continuity, so there is no overlap in our results our methodology.

Finally, in a very recent paper, Bécigneul and Ganea [5] established the convergence of a class of adaptive Riemannian methods in *geodesically* convex problems (extending in this way classical results for AdaGrad to a manifold setting). Importantly, the Riemannian methodology of [5] involves the exponential mapping of the underlying metric and focuses on geodesic convexity, so it concerns an orthogonal class of problems. The only overlap would be in the case of *flat* Riemannian manifolds:

however, even though the manifolds we consider here are topologically simple, they are *not* flat.[1] In view of this, there is no overlap with the analysis and results of [5].

## 2 PROBLEM SETUP

We begin by presenting the core online optimization framework that we will consider throughout the rest of our paper. This can be described by the following sequence of events:

1. At each round $t = 1, 2, \ldots$, the optimizer chooses an *action* $X_t$ from a convex – but not necessarily closed or compact – subset $\mathcal{X}$ of an ambient normed space $\mathcal{V} \cong \mathbb{R}^d$.
2. The optimizer incurs a loss $f_t(X_t)$ based on some (unknown) convex *loss function* $f_t \colon \mathcal{X} \to \mathbb{R}$.
3. The optimizer updates their action and the process repeats.

*Remark* 1. For posterity, we note that if $\mathcal{X}$ is not closed, $f_t$ (or its derivatives) could become singular at a residual point $x \in \mathrm{bd}(\mathcal{X}) \setminus \mathcal{X}$; in particular, we do not assume here that $f_t$ admits a smooth extension to the closure $\mathrm{cl}(\mathcal{X})$ of $\mathcal{X}$ (or even that it is bounded over bounded subsets of $\mathcal{X}$).

In this broad framework, the most widely used figure of merit is the minimization of the agent's regret. Formally, the *regret* of a policy $X_t \in \mathcal{X}$, $t = 1, 2, \ldots$, is defined as

$$\mathrm{Reg}_x(T) = \sum_{t=1}^{T} [f_t(X_t) - f_t(x)], \tag{1}$$

for all $x \in \mathcal{X}$. We then say that the policy $X_t$ leads to *no regret* if $\mathrm{Reg}_x(T) = o(T)$ for all $x \in \mathcal{X}$.

In addition to convexity, the standard assumption in the literature for the problem's loss functions is *Lipschitz continuity*, i.e.,

$$|f_t(x') - f_t(x)| \leq G_t \|x' - x\| \tag{LC}$$

for some $G_t \geq 0$, $t = 1, 2, \ldots$, and for all $x, x' \in \mathcal{X}$. Under (LC), if the agent observes at each stage $t$ an element $v_t$ of $\partial f_t(X_t)$, straightforward online policies based on gradient descent enjoy a bound of the form $\mathrm{Reg}_x(T) = \mathcal{O}(\bar{G}_T T^{1/2})$, with $\bar{G}_T^2 = T^{-1} \sum_{t=1}^{T} G_t^2$ [11, 35, 41]. In particular, if $G \equiv \limsup_{T \to \infty} \bar{G}_T < \infty$ (e.g., if each $f_t$ is $G$-Lipschitz continuous over $\mathcal{X}$), we have the bound

$$\mathrm{Reg}_x(T) = \mathcal{O}(G T^{1/2}) \tag{2}$$

which is well known to be min-max optimal in this setting [1].

**A note on notation.**   Throughout our paper, we make a clear distinction between $\mathcal{V}$ and its dual, and we use Dirac's notation $\langle v | x \rangle$ for the duality pairing between $v \in \mathcal{V}^*$ and $x \in \mathcal{V}$ (not to be confused with the notation $\langle \cdot, \cdot \rangle$ for a scalar product on $\mathcal{V}$). Also, unless mentioned otherwise, all notions of boundary and interior should be interpreted in the relative (as opposed to topological) sense. We also make the blanket assumption that the subdifferential $\partial f_t$ of $f_t$ admits a continuous selection $\nabla f_t(x) \in \partial f_t(x)$ for all $x \in \mathrm{dom}\, \partial f_t \equiv \{x \in \mathcal{X} : \partial f_t(x) \neq \varnothing\}$.

## 3 RIEMANN–LIPSCHITZ CONTINUITY

Despite its generality, (LC) may fail to hold in a wide range of problems and applications, ranging from support vector machines to Poisson inverse problems, quantum tomography, etc. [3, 8, 25]. The loss functions of these problems exhibit singularities at the boundary of the feasible region, so the standard regret analysis cited above no longer applies. Accordingly, our first step will be to introduce a family of *local* norms $\|\cdot\|_x$, $x \in \mathcal{X}$, such that a variant of (LC) holds even if the derivatives $\partial f / \partial x_i$ of $f$ blow up near the boundary of $\mathcal{X}$.

To achieve this, we will employ the notion of a *Riemannian metric*. This is simply a position-dependent scalar product on $\mathcal{V}$, i.e., a continuous assignment of bilinear pairings $\langle \cdot, \cdot \rangle_x$, $x \in \mathcal{X}$, satisfying the following conditions for all $z, z' \in \mathcal{V}$ and all $x \in \mathcal{X}$:

---

[1]For example, the open unit simplex endowed with the Shahshahani metric is isometric to the positive orthant of a sphere with the round metric (cf. Section 3). This space has constant positive curvature and the geodesics are portions of great circles, so the two analyses are very different in that case.

1. *Symmetry:* $\langle z, z' \rangle_x = \langle z', z \rangle_x$.

2. *Positive-definiteness:* $\langle z, z \rangle_x \geq 0$ with equality if and only if $z = 0$.

More concretely, in the standard basis $\{e_i\}_{i=1}^d$ of $\mathbb{R}^d$, we define the *metric tensor of* $\langle \cdot, \cdot \rangle_x$ as the matrix $g(x) \in \mathbb{R}^{d \times d}$ with components

$$g_{ij}(x) = \langle e_i, e_j \rangle_x \qquad i, j = 1, \ldots, d. \tag{3}$$

The norm of $z \in \mathcal{V}$ at $x \in \mathcal{X}$ is then defined as

$$\|z\|_x^2 \equiv \langle z, z \rangle_x^2 = \sum_{i,j=1}^d g_{ij}(x) z_i z_j = z^\top g(x) z. \tag{4}$$

In this way, a Riemannian metric allows us to measure lengths and angles between displacement vectors at each $x \in \mathcal{X}$; for illustration, we provide some notable examples below:

**Example 1** (Euclidean geometry). The ordinary *Euclidean metric* on $\mathcal{X} = \mathbb{R}^d$ is $g(x) = I$. This yields the standard expressions $\|z\|_x^2 = \sum_{i=1}^d z_i^2$ and $\langle z, z' \rangle_x = \sum_{i=1}^d z_i z_i'$, both independent of $x$.

**Example 2** (Hyperbolic geometry). The *Poincaré metric* on the positive orthant $\mathcal{X} = \mathbb{R}_{++}^d$ is

$$g(x) = \mathrm{diag}(1/x_1^2, \ldots, 1/x_d^2), \tag{5}$$

leading to the local norm $\|z\|_x^2 = \sum_{i=1}^d z_i^2 / x_i^2$. Under (5), $\mathbb{R}_{++}^d$ can be seen as a variant of Poincaré's half-space model for hyperbolic geometry [22]; this will become important later.

Given a Riemannian metric on $\mathcal{X}$, the length of a curve $\gamma \colon [0, 1] \to \mathcal{X}$ is defined as $L_g[\gamma] = \int_0^1 \|\dot{\gamma}(s)\|_{\gamma(s)} \, ds$, and the *Riemannian distance* between $x_1, x_2 \in \mathcal{X}$ is given by

$$\mathrm{dist}_g(x_1, x_2) = \inf_\gamma L_g[\gamma]. \tag{6}$$

Under this definition, it is natural to introduce the following Riemannian variant of (LC):

**Definition 1.** We say that $f \colon \mathcal{X} \to \mathbb{R}$ is *Riemann–Lipschitz continuous relative to $g$* if

$$|f(x') - f(x)| \leq G \, \mathrm{dist}_g(x, x') \quad \text{for some } G \geq 0 \text{ and all } x, x' \in \mathcal{X}. \tag{7}$$

Albeit simple to state, (RLC) may be difficult to verify because it requires the computation of the distance function $\mathrm{dist}_g$ of $g$ – which, in turn, relies on geodesic calculations to identify the shortest possible curve between two points. Nevertheless, if $f$ is differentiable, Proposition 1 below provides an alternative characterization of Riemann–Lipschitz continuity which is easier to work with:

**Proposition 1.** *Suppose that $f \colon \mathcal{X} \to \mathbb{R}$ is differentiable. Then, (RLC) holds if and only if*

$$\|\mathrm{grad}\, f(x)\|_x \leq G \quad \text{for all } x \in \mathcal{X}. \tag{RLC}$$

*Remark* 2. In the above, the *Riemannian gradient* $\mathrm{grad}\, f(x)$ of $f$ at $x$ is defined as follows: First, let $\mathcal{Z} = \mathrm{span}\{x' - x : x, x' \in \mathcal{X}\}$ denote the *tangent hull* of $\mathcal{X}$. Then, $\mathrm{grad}\, f(x) \in \mathcal{Z}$ is defined by the characteristic property

$$f'(x; z) = \langle \mathrm{grad}\, f(x), z \rangle_x \quad \text{for all } z \in \mathcal{Z}. \tag{8}$$

Existence and uniqueness of $\mathrm{grad}\, f(x)$ is due to the fact that $g(x)$ is positive-definite – and, hence, invertible [22]. In particular, if $\mathcal{X}$ is full-dimensional (so $\mathcal{Z} = \mathcal{V}$), we have:

$$[\mathrm{grad}\, f(x)]_i = \sum_{j=1}^d g(x)_{ij}^{-1} \partial_j f(x). \tag{9}$$

The proof of Proposition 1 requires the introduction of further tools from Riemannian geometry; seeing as these notions are not used anywhere else in our paper, we relegate it to the appendix. Instead, we close this section with a simple example of a singular function which is nonetheless Riemann–Lipschitz continuous:

**Example 3.** Let $\mathcal{X} = [0,1]^d \setminus \{0\}$ (so $\mathcal{X}$ is convex but neither open nor closed) and let $f(x) = -\log(a^\top x)$ for some positive vector $a \in \mathbb{R}^d_{++}$. If we take $g_{ij}(x) = \delta_{ij}/(x_1 + \cdots + x_d)^2$, we get

$$\|\operatorname{grad} f(x)\|_x^2 = \frac{\sum_{i=1}^d a_i^2 \cdot (\sum_{i=1}^d x_i)^2}{(a^\top x)^2} \leq \frac{\sum_{i=1}^d a_i^2}{(\min_j a_j)^2}. \tag{10}$$

Thus, although $f$ is not Lipschitz continuous in the standard sense, it *is* Riemann–Lipschitz continuous relative to $g$; we will revisit this example in our treatment of Poisson inverse problems in Section 6.

More generally, Example 3 suggests the following rule of thumb: if $f$ exhibits a gradient singularity of the form $|\partial_i f(x)| = \mathcal{O}(\phi(x))$ at some residual point $x \in \operatorname{cl}(\mathcal{X}) \setminus \mathcal{X}$ of $\mathcal{X}$, taking $g_{ij}(x) = \phi(x)^2 \delta_{ij}$ gives $\|\operatorname{grad} f(x)\|_x^2 = \phi(x)^{-2} \sum_{i=1}^d [\partial_i f(x)]^2 = \mathcal{O}(1)$. On that account, $f$ is Riemann–Lipschitz continuous, even though its derivative is singular; we find this heuristic particularly appealing because it provides a principled choice of Riemannian metric under which $f$ satisfies (RLC).

## 4 ALGORITHMS

In this section, we present the algorithms that we will study in the sequel: "follow the regularized leader" (FTRL) and online mirror descent (OMD). Both methods have been widely studied in the literature in the context of "vanilla" Lipschitz continuity; however, beyond this basic setting, treating FTRL/OMD in the Riemannian framework of the previous section is an intricate affair that requires several conceptual modifications. For this reason, we take an in-depth look into both methods below.[2]

### 4.1 REGULARIZATION

We begin with the idea of regularization through a suitable penalty function. In our Riemannian setting, we adapt this notion as follows:

**Definition 2.** Let $g$ be a Riemannian metric on $\mathcal{X}$ and let $h \colon \mathcal{V} \to \mathbb{R}$ be a proper lower semi-continuous (l.s.c.) convex function with $\operatorname{dom} h = \mathcal{X}$.[3] We say that $h$ is a *Riemannian regularizer* on $\mathcal{X}$ if:

1. The subdifferential of $h$ admits a *continuous selection*, i.e., a continuous function $\nabla h$ such that $\nabla h(x) \in \partial h(x)$ for all $x \in \mathcal{X}^\circ \equiv \operatorname{dom} \partial h$.

2. $h$ is *strongly convex relative to $g$*, i.e.,

$$h(x') \geq h(x) + \langle \nabla h(x) | x' - x \rangle + \tfrac{1}{2} K \|x' - x\|_x^2 \tag{11}$$

   for some $K > 0$ and all $x \in \mathcal{X}^\circ$, $x' \in \mathcal{X}$.

The *Bregman divergence* induced by $h$ is then defined for all $p \in \mathcal{X}$, $x \in \mathcal{X}^\circ$ as

$$D(p, x) = h(p) - h(x) - \langle \nabla h(x) | p - x \rangle. \tag{12}$$

There are two points worth noting in the above definition. First, the domain of $h$ is all of $\mathcal{X}$, but this need not be the case for the subdifferential $\partial h$ of $h$: by convex analysis arguments [33, Chap. 26], we have $\operatorname{ri} \mathcal{X} \subseteq \mathcal{X}^\circ \equiv \operatorname{dom} \partial h \subseteq \mathcal{X}$. To connect the two, we will say that $h$ is a *Riemann–Legendre regularizer* when $\mathcal{X}^\circ = \operatorname{ri} \mathcal{X}$ and $D(p, x_n) \to 0$ whenever $x_n \to p$.

Second, strong convexity in (11) is defined relative to the underlying Riemannian metric. If the norm in (11) does not depend on $x$, we recover the standard definition; however, the dependence of the second-order term in (11) on $g$ can change the landscape significantly. Lemma 1 and Example 5 below provide an illustration of this interplay between $g$ and $h$:

**Lemma 1.** *A Riemannian regularizer $h$ is $K$-strongly convex relative to $g$ if and only if*

$$D(p, x) \geq \tfrac{1}{2} K \|p - x\|_x^2. \tag{13}$$

---

[2]For convenience, we tacitly assume in what follows that $g(x) \succcurlyeq \mu I$ for some $\mu > 0$ and all $x \in \mathcal{X}$. Since $g \succcurlyeq 0$, this can always be achieved without loss of generality by replacing $g$ by $g + \mu I$.

[3]Following standard convex analysis terminology, "proper" means here that $h \not\equiv +\infty$ while lower semi-continuous refers to the property that $\liminf_{x \to x_0} h(x) \geq h(x_0)$ for all $x_0 \in \mathcal{V}$.

The proof of Lemma 1 follows from a rearrangement of (11) so we omit it. Instead, we present below some examples of Riemannian regularizers:

**Example 4.** Let $\mathcal{X} = [0, 1]^d$, and consider the so-called *Burg entropy* $h(x) = -\sum_{i=1}^d \log x_i$. It is easy to see that $h(x)$ is strongly convex relative to the standard Euclidean norm $\|\cdot\|_2$. Moreover, we have

$$D(p, x) = \sum_{i=1}^d \left[ \frac{p_i}{x_i} - \log \frac{p_i}{x_i} - 1 \right] \tag{14}$$

and, by Taylor's theorem with Lagrange remainder, we readily get

$$D(p, x) \geq \frac{1}{2} \sum_{i=1}^d \frac{(p_i - x_i)^2}{x_i} = \|p - x\|_x^2 \tag{15}$$

where $\|z\|_x = \sum_{i=1}^d z_i^2 / x_i$ denotes the so-called Shahshahani norm on $\mathcal{X}$ (i.e., $h$ is also strongly convex relative to $\|\cdot\|_x$). This regularizer has been used extensively in the setting of Poisson inverse problems and plays a central role in the analysis of Bauschke et al. [3], Hanzely and Richtárik [16], He et al. [18], and Lu et al. [25]. For completeness, we revisit it in Section 6.

**Example 5.** Let $\mathcal{X}$ and $g$ be as in Example 3, and let $h(x) = (1 + r^2)/\sum_{i=1}^d x_i$ with $r^2 = \sum_{i=1}^d x_i^2$. Then, a tedious (but otherwise straightforward) algebraic calculation gives

$$D(p, x) \geq \sum_{i=1}^d \frac{(x_i - p_i)^2}{(\sum_{j=1}^d x_j)^2} = \|p - x\|_x^2 \tag{16}$$

i.e., $h$ is strongly convex relative to $g$. By contrast, due to the singularity of $g$ at 0, it is easy to check that the Euclidean regularizer $h(x) = (1/2) \sum_{i=1}^d x_i^2$ *is not* strongly convex relative to $g$.

## 4.2 Algorithms and feedback structure

With these preliminaries in hand, we begin with the FTRL algorithm, which we state here as follows:

$$X_{t+1} = \arg\min_{x \in \mathcal{X}} \left\{ \sum_{s=1}^t f_s(x) + \gamma^{-1} h(x) \right\}. \tag{FTRL}$$

In the above, $\gamma > 0$ is a step-size parameter whose role is discussed below; as for the existence of the arg min, this is justified by the lower semicontinuity and strong convexity of $h$ together with the fact that $\text{dom } h = \mathcal{X}$ (so the minimum cannot be attained in the residual set $\text{cl}(\mathcal{X}) \setminus \mathcal{X}$ of $\mathcal{X}$).

In terms of feedback, FTRL assumes that the optimizer has access to all the loss functions encountered up to a given round (except, of course, for the current one). In many cases of practical interest, this assumption is too restrictive and, instead, the optimizer only has access to a first-order oracle for each $f_t$. To model this feedback structure, we assume that once $X_t$ has been chosen, the optimizer receives an estimate $v_t$ of $\nabla f_t(X_t)$ satisfying the following hypotheses:

a) *Unbiasedness:*     $\mathbb{E}[v_t \mid \mathcal{F}_t] = \nabla f_t(X_t).$ (17a)

b) *Finite mean square:*     $\mathbb{E}[\|v_t\|_*^2 \mid \mathcal{F}_t] \leq M_t^2.$ (17b)

In the above, $\|\cdot\|_*$ denotes the dual norm of $\|\cdot\|_{X_t}$, i.e., the Riemannian norm at the point $X_t$ where the oracle was called (we suppress here the index $X_t$ and write $\|\cdot\|_*$ instead of $\|\cdot\|_{X_t, *}$ to lighten the notation). In particular, the oracle feedback $v_t$ may fail to be bounded in $L^2$ relative to *any* global norm on $\mathcal{V}^*$; as such, (17) *is considerably weaker than the standard $L^2$-boundedness assumption for global norms.* Finally, in terms of measurability, the expectation in (17) is conditioned on the history $\mathcal{F}_t$ of $X_t$ up to stage $t$; since $v_t$ is generated randomly from $X_t$, it is *not* $\mathcal{F}_t$-measurable.

To proceed, the main idea of mirror descent is to replace $f_s(x)$ in (FTRL) with the first-order surrogate $f_s(x) \leftarrow f_s(X_s) + \langle \nabla f_s(X_s) | x - X_s \rangle$. In this way, substituting $v_s$ for the estimate of $\nabla f_s(X_s)$ received at stage $s$, we obtain the linearized FTRL scheme

$$X_{t+1} = \arg\min_{x \in \mathcal{X}} \{ \gamma \sum_{s=1}^t \langle v_s | x \rangle + h(x) \}. \tag{18}$$

To rewrite this process in recursive form, introduce the auxiliary (dual) variable

$$Y_{t+1} = Y_t - \gamma v_t \tag{19}$$

so $Y_{t+1} = -\gamma \sum_{s=1}^{t} v_s$, and hence

$$X_{t+1} = \arg\min_{x \in \mathcal{X}} \{h(x) - \langle Y_{t+1} | x \rangle\} = \arg\max_{x \in \mathcal{X}} \{\langle Y_{t+1} | x \rangle - h(x)\}. \tag{20}$$

Therefore, letting

$$Q(y) = \arg\max_{x \in \mathcal{X}} \{\langle y | x \rangle - h(x)\} \tag{21}$$

denote the so-called "mirror map" of the method, we obtain the following incarnation of the *online mirror descent* (OMD) algorithm:

$$\begin{aligned} Y_{t+1} &= Y_t - \gamma v_t \\ X_{t+1} &= Q(Y_{t+1}). \end{aligned} \tag{OMD}$$

This version of OMD is also known as "*dual averaging*" [26, 29, 30, 38] or "*lazy mirror descent*" [35]; for a "greedy" variant, see [6, 27, 28] and references therein.

## 5 ANALYSIS AND RESULTS

### 5.1 REGRET ANALYSIS

We begin by stating our main results for the regret minimization properties of FTRL and OMD. Throughout this section, we make the following blanket assumptions:

1. Both algorithms are initialized at the "prox-center" $x_c = \arg\min h$ of $\mathcal{X}$ and are run with (constant) step-size $\alpha/T^{1/2}$ for some $\alpha > 0$ chosen by the optimizer.

2. The $t$-th stage loss function $f_t \colon \mathcal{X} \to \mathbb{R}$ is convex and satisfies (RLC) with constant $G_t$.

3. The optimizer's aggregate loss $\sum_{t=1}^{T} f_t$ attains its minimum value at some $x^* \in \mathcal{X}$.

The purpose of the last assumption is to avoid cases where the infimum of a loss function is not attained within the problem's feasible region (such as $e^{-x}$ over $\mathbb{R}_+$). We then have:

**Theorem 1.** *Let* $\mathrm{Reg}(T) \equiv \mathrm{Reg}_{x^*}(T)$, $\bar{G}_T^2 = T^{-1} \sum_{t=1}^{T} G_t^2$, *and* $\bar{M}_T^2 = T^{-1} \sum_{t=1}^{T} M_t^2$. *Then:*

a) *The FTRL algorithm enjoys the regret bound*

$$\mathrm{Reg}(T) \leq \left[ \frac{D(x^*, x_c)}{\alpha} + \frac{2\alpha \bar{G}_T^2}{K} \right] \sqrt{T}. \tag{22a}$$

b) *The OMD algorithm with noisy feedback of the form* (17) *enjoys the mean regret bound*

$$\mathbb{E}[\mathrm{Reg}(T)] \leq \left[ \frac{D(x^*, x_c)}{\alpha} + \frac{\alpha \bar{M}_T^2}{2K} \right] \sqrt{T}. \tag{22b}$$

*In particular, if* $\sup_t G_t < \infty$, $\sup_t M_t < \infty$, *both algorithms guarantee* $\mathcal{O}(\sqrt{T})$ *regret.*

*Remark* 3. We emphasize here that the $\mathcal{O}(\sqrt{T})$ regret bound above is achieved even if $\mathcal{X}$ is unbounded or if the "Bregman depth" $H \equiv \sup_{x \in \mathcal{X}} D(x, x_c) = \sup h - \inf h$ of $\mathcal{X}$ is infinite.[4] Of course, if $H < \infty$ and $G$ (or $M$) is known to the optimizer, (22) can be optimized further by tuning $\alpha$. When these constants are unknown, achieving an optimized constant by means of an adaptive step-size policy is an important question, but one which lies beyond the scope of this paper.

The main idea behind the proof of Theorem 1 is to relate the Riemannian structure of $\mathcal{X}$ to the Bregman regularization framework underlying (FTRL) and (OMD). A first such link is provided by the Bregman divergence (12); however, because of the primal-dual interplay between $X_t \in \mathcal{X}$ and $Y_t \in \mathcal{V}^*$, the Bregman divergence is not sufficiently adapted. To overcome this difficulty, we employ the *Fenchel coupling* between a target point $p \in \mathcal{X}$ and $y \in \mathcal{V}^*$, defined here as

$$\Phi(p, y) = h(p) + h^*(y) - \langle y | p \rangle \quad \text{for all } p \in \mathcal{X}, y \in \mathcal{V}^*, \tag{23}$$

---

[4]To see this, simply note that $D(x, x_c) = h(x) - h(x_c) - \langle \nabla h(x_c) | x - x_c \rangle < \infty$ for all $x \in \mathcal{X} = \operatorname{dom} h$ (recall also that, since $x_c = \arg\min h$, we have $0 \in \partial h(x_c)$ so $x_c \in \operatorname{dom} \partial h$).

with $h^*(y) = \max_{x \in \mathcal{X}}\{\langle y|x\rangle - h(x)\}$ denoting the convex conjugate of $h$. As we show in the appendix, the Fenchel coupling (which is non-negative by virtue of Young's inequality) enjoys the key property

$$\Phi(p, y - \gamma v) \le \Phi(p, y) - \langle \gamma v | Q(y) - p \rangle + \frac{\gamma^2}{2K} \|v\|_{Q(y),*}^2. \tag{24}$$

It is precisely this primal-dual inequality which allows us to go beyond the standard Lipschitz framework: compared to (primal-primal) inequalities of a similar form for global norms [2, 21, 27, 30, 40], the distinguishing feature of (24) is the advent of the Riemannian norm $\|v\|_{x,*}$. Thanks to the intricate connection between $g$ and $h$, the second-order term in (24) can be controlled even when the received gradient is unbounded relative to *any* global norm, i.e., even if the objective is singular.

The main obstacle to achieve this is that the underlying Riemannian metric $g$, the Fenchel coupling $\Phi$ and the Bregman divergence $D$ (all state-dependent notions of distance) need not be compatible with one another. That this is indeed the case is owed to Lemma 1: tethering the Riemannian norm in (13) to the *second* argument of the Bregman divergence instead of the first (or any other point in-between) plays a crucial role in deriving (24). Any other relation between $g$ and $h$ along these lines is not amenable to analyzing (FTRL) or (OMD) in this framework.

## 5.2 Applications to stochastic optimization

The second part of our analysis concerns stochastic optimization problems of the form

$$\begin{aligned} \text{minimize} \quad & f(x) = \mathbb{E}[F(x; \omega)] \\ \text{subject to} \quad & x \in \mathcal{X} \end{aligned} \tag{Opt}$$

with the expectation taken over some model sample space $\Omega$. Our first result here is as follows:

**Theorem 2.** *Assume that $f$ is convex and Riemann–Lipschitz continuous in mean square, i.e., $\sup_x \mathbb{E}[\|\nabla F(x; \omega)\|_{x,*}^2] \le M^2$ for some $M > 0$. If (OMD) is run for $T$ iterations with a constant step-size of the form $\alpha/\sqrt{T}$ and stochastic gradients $v_t = \nabla F(X_t; \omega_t)$ generated by an i.i.d. sequence $\omega_t \in \Omega$, we have*

$$\mathbb{E}[f(\bar{X}_T)] \le \min f + \left[ \frac{D_c}{\alpha} + \frac{\alpha M^2}{2K} \right] \frac{1}{\sqrt{T}} \tag{25}$$

*where $\bar{X}_T = (1/T)\sum_{t=1}^T X_t$ is the "ergodic average" of $X_t$ and $D_c = \inf_{x^* \in \arg\min f} D(x^*, x_c) < \infty$ denotes the Bregman distance of the prox-center $x_c$ of $\mathcal{X}$ to $\arg\min f$.*

The key novelty in Theorem 2 is that the optimal $\mathcal{O}(T^{-1/2})$ convergence rate of OMD is maintained *even if the stochastic gradients of $F$ become singular at residual points $x \in \mathrm{cl}(\mathcal{X}) \setminus \mathcal{X}$*. As with the regret guarantee of Theorem 1, this is achieved by the intricate three-way relation between the landscape of $f$, the underlying Riemannian metric $g$ (which is tailored to the singularity profile of the latter), and the Riemannian regularizer $h$. The proof of Theorem 2 likewise relies on an online-to-batch conversion of the regret guarantees of (OMD) for the sequence of stochastic gradients $\nabla F(\cdot; \omega_t)$ of $f$; the details can be found in the appendix.

To go beyond the ergodic guarantees of Theorem 2, we also analyze below the convergence of the "last iterate" of OMD, i.e., the *actual sequence of generated points $X_t$*. This is of particular interest for non-convex problems where ergodic convergence results are of limited value (because Jensen's inequality no longer applies). To obtain global convergence results in this setting, we focus on a class of functions which satisfy a weak secant inequality of the form

$$\inf\{\langle \nabla f(x)|x - x^*\rangle : x^* \in \arg\min f, x \in \mathcal{K}\} > 0 \tag{SI}$$

for every closed subset $\mathcal{K}$ of $\mathcal{X}$ that is separated by neighborhoods from $\arg\min f$. Variants of this condition have been widely studied in the literature and include non-convex functions with complicated ridge structures [9, 13, 19, 20, 23, 31, 39, 40]. In this very general setting, we have:

**Theorem 3.** *Assume $f$ satisfies (SI) and is Riemann–Lipschitz continuous in $L^2$. Suppose further that $\arg\min f$ is bounded and (OMD) is run with a sequence of stochastic gradients $v_t = \nabla F(X_t; \omega_t)$, a Riemann–Legendre regularizer $h$, and a variable step-size $\gamma_t$ such that $\sum_{t=1}^\infty \gamma_t = \infty$, $\sum_{t=1}^\infty \gamma_t^2 < \infty$. Then, with probability 1, $X_t$ converges to some (possibly random) $x^* \in \arg\min f$.*

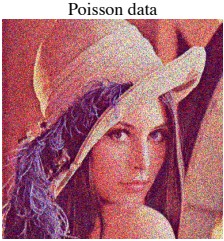 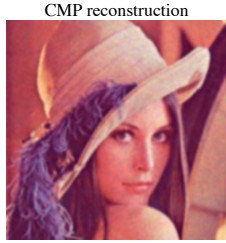 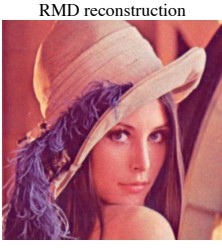 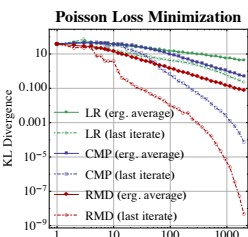

**Figure 1:** Reconstruction of the Lena test image from a sample contaminated with Poisson noise. Left to right: (*a*) the contaminated sample; (*b*) CMP reconstruction; (*c*) RMD reconstruction; and (*d*) Poisson likelihood loss at each iteration. The RMD process provides a sharper definition of image features relative to the CMP algorithm (which is the second-best).

The proof for Theorem 3 hinges on combining (quasi-)supermartingale convergence results with the basic inequality (24); we detail the proof in the paper's appendix. Seeing as (SI) holds trivially for (pseudo-)convex functions, we only note here that Theorem 3 complements Theorem 2 in an important way: the convergence of $X_t$ implies that of $\bar{X}_t$, so the convergence of $\mathbb{E}[f(\bar{X}_t)]$ to min $f$ follows immediately from Theorem 3; however, the rate of convergence (25) doesn't. In practice, the ergodic average converges to interior minimizers faster than the last iterate but lags behind when tracking boundary points and/or in non-convex landscapes; we explore this issue in Section 6 below.

## 6 NUMERICAL EXPERIMENTS IN POISSON INVERSE PROBLEMS

For the purposes of validation, we proceed with an application of our algorithmic results to a broad class of Poisson inverse problems that arise in tomography problems. Referring the reader to the appendix for the details, the objective of interest here is the Poisson likelihood loss (generalized Kullback–Leibler divergence):

$$f(x) = \sum_{j=1}^{m} \left[ u_j \log \frac{u_j}{(Hx)_j} + (Hx)_j - u_j \right] \tag{26}$$

where $u \in \mathbb{R}_+^m$ is a vector of Poisson data observations (e.g., pixel intensities) and $H \in \mathbb{R}^{m \times d}$ is an ill-conditioned matrix representing the data-gathering protocol. Since the generalized KL objective of (26) exhibits an $\mathcal{O}(1/x)$ singularity at the boundary of the orthant, we consider the Poincaré metric $g(x) = \mathrm{diag}(1/x_1, \dots, 1/x_d)$ under which the KL divergence is Riemann–Lipschitz continuous (cf. Example 2). Going back to Example 3, a suitable Riemannian regularizer for this metric is $h(x) = \sum_{i=1}^{m} 1/x_i^2$, which is 1-strongly convex relative to $g$. We then run the induced mirror descent algorithm with an online-to-batch conversion mechanism as described in Section 5.2. For reference purposes, we call the resulting process *Riemannian mirror descent* (RMD).

Subsequently, we ran RMD on a Poisson denoising problem for a $384 \times 384$ test image contaminated with Poisson noise (so $d \approx 10^5$ in this case). For benchmarking, we also ran a fast variant of the widely used Lucy–Richardson (LR) algorithm [7], and the recent composite mirror prox (CMP) method of [18]; all methods were run with stochastic gradients and the same minibatch size. Because of the "dark area" gradient singularities when $[Hx]_j \to 0$, Euclidean stochastic gradient methods oscillate without converging, so they are not reported. As we see in Fig. 1, the RMD process provides the sharpest reconstruction of the original. In particular, after an initial warm-up phase, the last iterate of Riemannian mirror descent consistently outperforms the LR algorithm by 7 orders of magnitude, and CMP by 3. We also note that the Poisson likelihood loss decreases faster under the last iterate of RMD relative to the different algorithmic variants that we tested, exactly because of the hysteresis effect that is inherent to ergodic averaging.

Overall, we note that the introduction of an additional degree of freedom (the choice of Bregman function and that of the local Riemannian norm), makes RMD a particularly flexible and powerful paradigm for loss models with singularities. We find these results particularly encouraging for further investigations on the interplay between Riemannian geometry and Bregman-proximal methods.

## 7 Concluding remarks

Owing to its connections with machine learning (support vector machines, Poisson inverse problems, quantum tomography, etc.), venturing beyond Lipschitz continuity is a fruitful research direction that has recently generated considerable interest in the literature. Depending on the type of continuity or smoothness encountered (Lipschitz continuity of the objective or Lipschitz continuity of the objective's gradients), the results can be significantly different, and it is not a priori clear which surrogate smoothness/continuity condition would be the most appropriate for any given problem. The present paper provides a complementary, Riemannian-geometric viewpoint which we feel can be fairly promising for the design of efficient optimization algorithms in this context. The precise characterization of the interplay between the different continuity conditions considered in the literature is an important open issue which we leave for future work.

### Acknowledgments

The authors gratefully acknowledge financial support from the French National Research Agency (ANR) under grants ORACLESS (ANR–16–CE33–0004–01) and ELIOT (ANR-18-CE40-0030), as well as the FAPESP 2018/12579-7 project. This work also benefited from financial support by MIAI Grenoble Alpes (Multidisciplinary Institute in Artificial Intelligence).

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

## A   RIEMANN–LIPSCHITZ CONTINUITY

In this appendix, our main goal is to prove Proposition 1, i.e., the equivalence between (7) and (RLC) when $f$ is differentiable.

To do so, we first need to introduce the notion of a *geodesic*, i.e., a length-minimizing curve that attains the infimum $\inf_\gamma L[\gamma]$ over all piecewise smooth curves joining two points $x_1, x_2 \in \mathcal{U}$. That such a curve exists and is unique in our setting is a basic fact of Riemannian geometry [22]. Moreover, given a tangent vector $z \in \mathcal{Z}$, this leads to the definition of the *exponential mapping* $\exp\colon \mathcal{U} \times \mathcal{Z} \to \mathcal{U}$ so that

$$(x, z) \mapsto \exp_x(z) = \gamma_z(1), \tag{A.1}$$

where $\gamma_z$ denotes the unique geodesic emanating from $x$ with initial velocity vector $\dot{\gamma}_z = z$. We then have $\exp_x(tz) = \gamma_z(t)$ for all $t$ and, moreover, for sufficiently small $r > 0$, the restriction of $\exp_x$ to a ball of radius $r$ in $\mathcal{Z}$ is a diffeomorphism onto its image in $\mathcal{U}$. The largest positive number $i_x$ such that the above holds for all $r < i_x$ is then known as the injectivity radius of $\mathcal{U}$ at $x$ [22].

Our proof of the equivalence between (7) and (RLC) follows a simplified version of the approach of Ferreira [14] who, to our knowledge, was the first to discuss the concept of proximal subgradients in Riemannian manifolds. To that end, fix some $x \in \mathcal{U}$, let $z = \operatorname{grad} f(x)$ and consider the *ray* emanating from $x$

$$\gamma(t) = \exp_x(tz/\|z\|_x). \tag{A.2}$$

Since $\gamma$ is a geodesic, we readily obtain $\operatorname{dist}_g(x, \gamma(t)) = t$ for all sufficiently small $t$. Also, by construction, we have $\exp_x^{-1} \gamma(t) = tz/\|z\|_x$. Hence, with $f$ convex, it follows that, for some constant $\alpha > 0$ and for sufficiently small positive $\delta < i_x$, we have

$$f(\gamma(t)) - f(x) \geq \langle z, \exp_x^{-1}(\gamma(t))\rangle \tag{A.3}$$
$$- \alpha \operatorname{dist}_g(x, \gamma(t))^2, \tag{A.4}$$

where we used the local topological equivalence of the Riemannian topology and the standard topology of $\mathbb{R}^d$, and the fact that $\exp_z(t)$ is a diffeomorphism for sufficiently small $\delta > 0$ – and hence, for all $\delta < i_x$. Thus, if $f$ is also Riemann–Lipschitz continuous in the sense of (7), we will also have

$$Gt \geq f(\gamma(t)) - f(x) \geq \langle z, tz/\|z\|_x\rangle_x - \alpha t^2. \tag{A.5}$$

Thus, by isolating the leftmost and rightmost hand sides, dividing by $t$, and taking the limit $t \to 0$, we get

$$\|\operatorname{grad} f(x)\|_x = \|z\|_x \leq G, \tag{A.6}$$

as was to be shown.

To establish the converse, assume that (RLC) holds, fix $x, x' \in \mathcal{X}$ pick $K > G$ and a sufficiently small $\delta > 0$, and consider the Riemannian distance majorant $w(x) = \operatorname{dist}_g(x, x')$ if $\operatorname{dist}_g(x, x') < \delta$, and $w(x) = K \operatorname{dist}_g(x, x') + \varepsilon^2/(\delta - \varepsilon)$ when $\delta < \operatorname{dist}_g(x, x') < 2\delta$, with $\varepsilon = \operatorname{dist}_g(x, x') - \delta$.

A simple calculation then shows that $\|\operatorname{grad} w(x)\|_x \geq K > G$. It is also straightforward to show that the minimum of $f + w$ is attained at $x'$, so

$$\begin{aligned} f(x') &= f(x') + w(x') \\ &\leq f(x) + w(x) \\ &\leq f(x) + K \operatorname{dist}_g(x, x'). \end{aligned} \tag{A.7}$$

Then, by interchanging $x$ and $x'$ above, we obtain $|f(x) - f(x')| \leq K \operatorname{dist}_g(x, x')$. Since $K > G$ has been chosen arbitrarily, (7) follows.

## B   PROPERTIES OF MIRROR MAPPINGS AND THE FENCHEL COUPLING

We begin by recalling and clarifying some of the notational conventions used in the paper. First, let $\mathcal{V} \cong \mathbb{R}^d$ be a finite-dimensional real space; then, its dual space will be denoted by $\mathcal{Y} \equiv \mathcal{V}^*$, and we write $\langle y|x\rangle$ for the duality pairing between $y \in \mathcal{Y}$ and $x \in \mathcal{V}$. Also, if $\|\cdot\|$ is a norm on $\mathcal{V}$, the dual norm on $\mathcal{Y}$ is defined as $\|y\|_* \equiv \sup\{\langle y|x\rangle : \|x\| \leq 1\}$.

Given an extended-real-valued convex function $f\colon \mathcal{X} \to \mathbb{R} \cup \{\infty\}$, we will write $\operatorname{dom} f \equiv \{x \in \mathcal{V} : f(x) < \infty\}$ for its *effective domain*. The *subdifferential* of $f$ at $x \in \operatorname{dom} f$ is then defined as $\partial f(x) \equiv \{y \in \mathcal{Y} : f(x') - f(x) + \langle y|x' - x\rangle \geq 0 \text{ for all } x' \in \mathcal{V}\}$ and the *domain of subdifferentiability* of $f$ is $\operatorname{dom} \partial f \equiv \{x \in \operatorname{dom} f : \partial f \neq \varnothing\}$. Finally, assuming it exists, the *directional derivative* of $f$ at $x$ along $z \in \mathcal{V}$ is defined as $f'(x; z) \equiv d/dt|_{t=0} f(x + tz)$. We will then say that $f$ is *differentiable* at $x \in \operatorname{dom} f$ if there exists $\nabla f(x) \in \mathcal{Y}$ such that $\langle \nabla f(x)|z\rangle = f'(x; z)$ for all vectors of the form $z = x' - x$, $x' \in \operatorname{dom} f$.

With these notational conventions at hand, we proceed to prove some auxiliary results and estimates that are used throughout the analysis of Section 5. To recall the basic setup, we assume throughout what follows that $h$ is a Riemannian regularizer in the sense of Definition 2. The convex conjugate $h^*\colon \mathcal{Y} \to \mathbb{R}$ of $h$ is then defined as

$$h^*(y) = \sup_{x \in \mathcal{X}} \{\langle y|x\rangle - h(x)\}. \tag{B.1}$$

Since $h$ is $K$-strongly convex relative to $g$, it is also strongly convex relative to the Euclidean norm (recall here that $g(x) \succcurlyeq \mu I$). As a result, the supremum in (B.1) is always attained, and $h^*(y)$ is finite for all $y \in \mathcal{Y}$ [4]. Moreover, by standard results in convex analysis [33, Chap. 26], $h^*$ is differentiable on $\mathcal{Y}$ and its gradient satisfies the identity

$$\nabla h^*(y) = \arg\max_{x \in \mathcal{X}} \{\langle y|x\rangle - h(x)\}. \tag{B.2}$$

Thus, recalling the definition of the mirror map $Q\colon \mathcal{Y} \to \mathcal{X}$ (cf.. Section 4):

$$Q(y) = \arg\max_{x \in \mathcal{X}} \{\langle y|x\rangle - h(x)\}, \tag{B.3}$$

we readily get

$$Q(y) = \nabla h^*(y). \tag{B.4}$$

Together with the prox-mapping induced by $h$, all these notions are related as follows:

**Lemma B.1.** *Let $h$ be a Riemannian regularizer on $\mathcal{X}$. Then, for all $x \in \operatorname{dom} \partial h$ and all $y, v \in \mathcal{Y}$, we have:*

$$a) \quad x = Q(y) \qquad \Longleftrightarrow \quad y \in \partial h(x). \tag{B.5a}$$

$$b) \quad x^+ = Q(\nabla h(x) + v) \iff \nabla h(x) + v \in \partial h(x^+) \tag{B.5b}$$

*Finally, if $x = Q(y)$ and $p \in \mathcal{X}$, we have*

$$\langle \nabla h(x)|x - p\rangle \leq \langle y|x - p\rangle. \tag{B.6}$$

*Remark.* Note that (B.5b) directly implies that $\partial h(x^+) \neq \varnothing$, i.e., $x^+ \in \operatorname{dom} \partial h$ for all $v \in \mathcal{Y}$. An immediate consequence of this is that the update rule $x^+ = Q(\nabla h(x) + v)$ is *well-posed*, i.e., it can be iterated in perpetuity.

*Proof of Lemma B.1.* To prove (B.5a), note that $x$ solves (B.2) if and only if $y - \partial h(x) \ni 0$, i.e., if and only if $y \in \partial h(x)$. Eq. (B.5b) is then obtained in the same manner.

For the inequality (B.6), it suffices to show it holds for all $p \in \mathcal{X}° \equiv \operatorname{dom} \partial h$ (by continuity). To do so, let

$$\phi(t) = h(x + t(p - x)) - [h(x) + \langle y|x + t(p - x)\rangle]. \tag{B.7}$$

Since $h$ is strongly convex relative to $g$ and $y \in \partial h(x)$ by (B.5a), it follows that $\phi(t) \geq 0$ with equality if and only if $t = 0$. Moreover, note that $\psi(t) = \langle \nabla h(x + t(p - x)) - y|p - x\rangle$ is a continuous selection of subgradients of $\phi$. Given that $\phi$ and $\psi$ are both continuous on $[0, 1]$, it follows that $\phi$ is continuously differentiable and $\phi' = \psi$ on $[0, 1]$. Thus, with $\phi$ convex and $\phi(t) \geq 0 = \phi(0)$ for all $t \in [0, 1]$, we conclude that $\phi'(0) = \langle \nabla h(x) - y|p - x\rangle \geq 0$, from which our claim follows. $\blacksquare$

As we mentioned earlier, much of our analysis revolves around a "primal-dual" divergence between a target point $p \in \mathcal{X}$ and a dual vector $y \in \mathcal{Y}$, called the *Fenchel coupling*. Following [26], this is defined as follows for all $p \in \mathcal{X}, y \in \mathcal{Y}$:

$$\Phi(p, y) = h(p) + h^*(y) - \langle y|p\rangle. \tag{B.8}$$

The following lemma illustrates some basic properties of the Fenchel coupling:

**Lemma B.2.** *Let $h$ be a Riemannian regularizer on $\mathcal{X}$ with convexity modulus $K$. Then, for all $p \in \mathcal{X}$ and all $y \in \mathcal{Y}$, we have:*

1. *$\Phi(p, y) = D(p, Q(y))$ if $Q(y) \in \mathcal{X}^\circ$ (but not necessarily otherwise).*

2. *If $x = Q(y)$, then $\Phi(p, y) \geq \frac{K}{2} \|x - p\|_x^2$*

*Proof.* For our first claim, let $x = Q(y)$. Then, by definition we have:

$$\Phi(p, y) = h(p) - \langle y|Q(y)\rangle - h(Q(y)) - \langle y|p\rangle = h(p) - h(x) - \langle y|p - x\rangle. \tag{B.9}$$

Since $y \in \partial h(x)$, we have $h'(x; p - x) = \langle y|p - x\rangle$ whenever $x \in \mathcal{X}^\circ$, thus proving our first claim. For our second claim, working in the previous spirit we get that:

$$\Phi(p, y) = h(p) - h(x) - \langle y|p - x\rangle \tag{B.10}$$

Thus, we obtain the result by recalling the strong convexity assumption for $h$ with respect to the Riemannian norm $\|\cdot\|_x$. ∎

We continue with some basic relations connecting the Fenchel coupling relative to a target point before and after a gradient step. The basic ingredient for this is a primal-dual analogue of the so-called "three-point identity" for Bregman functions [12]:

**Lemma B.3.** *Let $h$ be a regularizer on $\mathcal{X}$. Fix some $p \in \mathcal{X}$ and let $y, y^+ \in \mathcal{Y}$. Then, letting $x = Q(y)$, we have*

$$\Phi(p, y^+) = \Phi(p, y) + \Phi(x, y^+) + \langle y^+ - y|x - p\rangle. \tag{B.11}$$

*Proof.* By definition, we get:

$$\begin{aligned}
\Phi(p, y^+) &= h(p) + h^*(y^+) - \langle y^+|p\rangle \\
\Phi(p, y) &= h(p) + h^*(y) - \langle y|p\rangle.
\end{aligned} \tag{B.12}$$

Then, by subtracting the above we get:

$$\begin{aligned}
\Phi(p, y^+) - \Phi(p, y) &= h(p) + h^*(y^+) - \langle y^+|p\rangle - h(p) - h^*(y) + \langle y|p\rangle \\
&= h^*(y^+) - h^*(y) - \langle y^+ - y|p\rangle \\
&= h^*(y^+) - \langle y|Q(y)\rangle + h(Q(y)) - \langle y^+ - y|p\rangle \\
&= h^*(y^+) - \langle y|x\rangle + h(x) - \langle y^+ - y|p\rangle \\
&= h^*(y^+) + \langle y^+ - y|x\rangle - \langle y^+|x\rangle + h(x) - \langle y^+ - y|p\rangle \\
&= \Phi(x, y^+) + \langle y^+ - y|x - p\rangle
\end{aligned} \tag{B.13}$$

and our proof is complete. ∎

With all this at hand, we have the following key estimate:

**Proposition B.1.** *Let $h$ be a Riemannian regularizer on $\mathcal{X}$ with convexity modulus $K$, fix some $p \in \mathcal{X}$, let $x = Q(y)$ for some $y \in \mathcal{Y}$. Then, for all $v \in \mathcal{Y}$, we have:*

$$\Phi(p, y + v) \leq \Phi(p, y) + \langle v|x - p\rangle + \frac{1}{2K}\|v\|_{x,*}^2 \tag{B.14}$$

*Proof.* By the three-point identity (B.11), we get

$$\Phi(p, y) = \Phi(p, y + v) + \Phi(Q(y + v), y) + \langle y - (y + v)|Q(y + v) - p\rangle \tag{B.15}$$

and hence, after rearranging:

$$\begin{aligned}
\Phi(p, y + v) &= \Phi(p, y) - \Phi(Q(y + v), y) + \langle v|Q(y + v) - p\rangle \\
&= \Phi(p, y) - \Phi(Q(y + v), y) + \langle v|x - p\rangle + \langle v|Q(y + v) - x\rangle
\end{aligned} \tag{B.16}$$

By Young's inequality [33], we also have

$$\langle v|Q(y + v) - x\rangle \leq \frac{K}{2}\|Q(y + v) - x\|_x^2 + \frac{1}{2K}\|v\|_{x,*}^2 \tag{B.17}$$

Our claim then follows by the fact that $\Phi(Q(y + v), y) \geq \frac{K}{2}\|Q(y + v) - x\|_x^2$ (cf. Lemmas 1 and B.2). ∎

## C   ANALYSIS OF FTRL

Our goal here is to prove the regret bound (22a) of Theorem 1. The starting point of our analysis is the following basic bound:

**Lemma C.1** (35, Lemma 2.3). *The sequence of actions generated by* (FTRL) *satisfies*

$$\text{Reg}_x(T) \le h(x) - h(X_1) + \sum_{t=1}^{T} [f_t(X_t) - f_t(X_{t+1})].$$

(C.1)

Importantly, the above bound does not require any Lipschitz continuity or strong convexity assumptions, so it applies to our setting "as is". The importance of Riemann–Lipschitz continuity lies in the following:

**Lemma C.2.** *If $f$ is convex and Riemann–Lipschitz continuous with constant $G$, then:*

$$f(x) - f(x') \le G\|x' - x\|_x \quad \text{for all } x, x' \in \mathcal{X}^{\circ}.$$

(C.2)

*Proof.* By the convexity of $f$, we have:

$$\begin{aligned}
f(x) - f(x') &\le \langle \nabla f(x) | x - x' \rangle = \langle \text{grad } f(x), x - x' \rangle_x \\
&\le \|\text{grad } f(x)\|_x \|x - x'\|_x \\
&\le G\|x - x'\|_x
\end{aligned}$$

(C.3)

where the first line follows from the definition of the Riemannian gradient of $f$, the second one from the Cauchy-Schwartz inequality, and the last from Proposition 1. ∎

With these preliminary results at hand, we obtain the following basic bound for FTRL:

**Proposition C.1.** *Suppose that* (FTRL) *is run against a sequence of loss convex loss functions with assumptions as in Section 5. Then, for all $x \in \mathcal{X}$, we have:*

$$\text{Reg}_x(T) \le \frac{D(x, x_c)}{\gamma} + \frac{2\gamma}{K} \sum_{t=1}^{T} G_t^2$$

(C.4)

*Proof.* Our proof is patterned after Shalev-Shwartz [35], but with an important difference regarding the use of Riemannian norms and Riemann–Lipschitz continuity. To begin, let

$$F_t(x) = \sum_{s=1}^{t-1} f_s(x) + \frac{h(x)}{\gamma}$$

(C.5)

denote the "cumulative" loss faced by the optimizer up to roun $t - 1$, including the regularization penalty. By the definition of the FTRL policy, $X_t \in \arg\min F_t(x)$, so $\langle \nabla F_t(X_t) | x - X_t \rangle \ge 0$ and, likewise, $\langle \nabla F_{t+1}(X_{t+1}) | x - X_{t+1} \rangle \ge 0$ for all $x \in \mathcal{X}$. Furthermore, since $h$ is $K$-strongly convex relative to $g$, $F_t$ and $F_{t+1}$ will be $(K/\gamma)$-strongly convex relative to $g$.

Putting all this together, we obtain:

$$F_t(X_{t+1}) \ge F_t(X_t) + \frac{K}{2\gamma} \|X_{t+1} - X_t\|_{X_{t+1}}^2$$

(C.6a)

$$F_{t+1}(X_t) \ge F_{t+1}(X_{t+1}) + \frac{K}{2\gamma} \|X_t - X_{t+1}\|_{X_t}^2$$

(C.6b)

and hence, after summing the above inequalities:

$$\begin{aligned}
f_t(X_t) - f_t(X_{t+1}) &\ge \frac{K}{2\gamma} \|X_{t+1} - X_t\|_{X_{t+1}}^2 + \frac{K}{2\gamma} \|X_t - X_{t+1}\|_{X_t}^2 \\
&\ge \frac{K}{2\gamma} \|X_{t+1} - X_t\|_{X_t}^2.
\end{aligned}$$

(C.7)

On the other hand, Lemma C.2 gives

$$f_t(X_t) - f_t(X_{t+1}) \le G_t \|X_t - X_{t+1}\|_{X_t}$$

(C.8)

so, combining the last two inequalities, we get:

$$\frac{K}{2\gamma}\|X_{t+1} - X_t\|_{X_t} \le G_t. \tag{C.9}$$

Therefore, plugging this back into (C.8) yields

$$f_t(X_t) - f_t(X_{t+1}) \le \frac{2\gamma G_t^2}{K}, \tag{C.10}$$

and our result obtains from Lemma C.1. ∎

The proof of (22a) then follows by applying Proposition C.1 with a step-size of the prescribed form.

## D ERGODIC ANALYSIS OF OMD

This appendix is devoted to the proof of our main regret bound for (OMD). We begin by recalling ther recursive definition of the (lazy) OMD method:

$$\begin{aligned} Y_{t+1} &= Y_t - \gamma v_t \\ X_{t+1} &= Q(Y_{t+1}) \end{aligned} \tag{OMD}$$

with $Q$ defined as in Appendix B and oracle feedback subject to the hypotheses (17). We may then write the oracle feedback received by the optimizer at time $t$ as $v_t = \nabla f_t(X_t) + U_{t+1}$; hence, by the unbiasedness assumption (17a), it follows that $\mathbb{E}[U_{t+1} \mid \mathcal{F}_t] = 0$, i.e., $U_t$ is a martingale difference sequence (MDS) relative to $\mathcal{F}_t$.

Now, applying Proposition B.1 to (OMD), we get:

$$\begin{aligned} \Phi(x^*, Y_{t+1}) &\le \Phi(x^*, Y_t) - \gamma\langle v_t | X_t - x^* \rangle + \frac{\gamma^2}{2K}\|v_t\|_{X_t,*}^2 \\ &= \Phi(x^*, Y_t) + \gamma\langle \nabla f_t(X_t) | x^* - X_t \rangle - \gamma\langle U_{t+1} | X_t - x^* \rangle + \frac{\gamma^2}{2K}\|v_t\|_{X_t,*}^2. \end{aligned} \tag{D.1}$$

Hence, after rearranging and telescoping, we obtain

$$\text{Reg}(T) \le \sum_{t=1}^T \langle \nabla f_t(X_t) | X_t - x^* \rangle \le \frac{D(x^*, x_c)}{\gamma} + \sum_{t=1}^T \xi_{t+1} + \frac{\gamma}{2K} \sum_{t=1}^T \|v_t\|_{X_t,*}^2 \tag{D.2}$$

where, in the last line, we used the definition of the Riemannian dual norm $\|\cdot\|_* \equiv \|\cdot\|_{x^*,*}$, and we set $\xi_{t+1} = \langle U_{t+1} | x^* - X_t \rangle$. Our result then follows by taking expectations on both sides.

## E LAST-ITERATE ANALYSIS OF OMD

In this last section, we will present the convergence analysis for the last iterate of (OMD) to $\arg\min f$.

We begin by recalling two important results from probability theory. The first is a version of the law of large numbers for martingale difference sequences that are bounded in $L^2$ [15]:

**Theorem E.1.** *Let $Y_t = \sum_{i=1}^t \zeta_i$ be a martingale and $\beta_t$ a non-decreasing positive sequence such that $\lim_{t\to\infty} \beta_t = \infty$. Then,*

$$\lim_{t\to\infty} Y_t/\beta_t = 0 \quad \text{almost surely} \tag{E.1}$$

*on the set $\sum_{t=1}^\infty \beta_t^{-2} \mathbb{E}[\zeta_t^2 \mid \mathcal{F}_{t-1}] < \infty$.*

The second is a convergence result for quasi-supermartingales due to Robbins and Sigmund [32]:

**Lemma E.1.** *Let $(\mathcal{F}_t)_{t\in\mathbb{N}}$ be a non-decreasing sequence of $\sigma-$ algebras. Let $(\alpha_t)_{t\in\mathbb{N}}$, $(\theta_t)_{t\in\mathbb{N}}$ non-negative $\mathcal{F}_t-$ measurable random variables, $(\eta_t)_{t\in\mathbb{N}}$ is an $\mathcal{F}_t-$ measurable non-negative summable random variable and the following inequality holds:*

$$\mathbb{E}[\alpha_{t+1} \mid \mathcal{F}_t] \le \alpha_t - \theta_t + \eta_t \quad \text{almost surely} \tag{E.2}$$

*Then, $(\alpha_t)_{t\in\mathbb{N}}$ converges almost surely towards a $[0, \infty)$-valued random variable.*

An application of this lemma leads us to the following result which is of independent interest:

**Proposition E.1.** *Let $X_t$ be the sequence of iterates generated by* (OMD) *run with a step-size sequence $\gamma_t$ such that $\sum_{t=1}^{\infty} \gamma_t^2 < \infty$ and a stochastic oracle as in the statement of Theorems 2 and 3. Then, for all $x^* \in \arg\min f$, $\Phi(x^*, Y_t)$ converges with probability* 1.

*Proof.* Let $x^* \in \arg\min f$. Recalling our main estimation:

$$\Phi(x^*, Y_{t+1}) \leq \Phi(x^*, Y_t) - \gamma_t \langle v_t | X_t - x^* \rangle_x + \frac{\gamma_t^2}{2K} \|v_t\|_{X_t,*}^2 \tag{E.3}$$

and taking conditional expectations on both sides, we get due to $\mathcal{F}_t-$ measurability arguments:

$$\mathbb{E}[\Phi(x^*, Y_{t+1})|\mathcal{F}_t] \leq \Phi(x^*, Y_t) - \gamma_t \langle v_t | X_t - x^* \rangle_x + \frac{\gamma_t^2}{2K} \mathbb{E}[\|v_t\|_{X_t,*}^2|\mathcal{F}_t]. \tag{E.4}$$

Since, $(2K)^{-1} \sum_{t=1}^{\infty} \gamma_t^2 \mathbb{E}[\|v_t\|_{X_t,*}^2|\mathcal{F}_t] \leq M(2K)^{-1} \sum_{t=1}^{\infty} \gamma_t^2 < \infty$. Thus, by applying the above we get the result. ∎

Having this at hand, we can establish the following proposition:

**Proposition E.2.** *Let $X_t$ be the sequence of iterates generated by* (OMD) *with assumptions as in Theorem 3. Then, for all $x^* \in \arg\min f$, the sequence $\|X_t - x^*\|_{X_t}$ is bounded with probability* 1.

*Proof.* Recalling our main estimation and taking condition expectations on both sides, we get:

$$\mathbb{E}[\Phi(x^*, Y_{t+1}) \mid \mathcal{F}_t] \leq \Phi(x^*, Y_t) - \gamma_t \langle v_t | X_t - x^* \rangle_x + \frac{\gamma_t^2}{2K} \mathbb{E}[\|v_t\|_{X_t,*}^2|\mathcal{F}_t] \tag{E.5}$$

Hence, by the above corollary, we have that the sequence $\Phi(x^*, Y_t)$ converges with probability 1 for all $x^* \in \arg\min f$. Thus, it is also bounded with probability 1 for all $x^*$. We then get

$$\|X_t - x^*\|_{X_t}^2 \leq \frac{2}{K} \Phi(x^*, Y_t) \tag{E.6}$$

which concludes our proof. ∎

We continue by showing that $X_t$ possesses a subsequence that converges to $\arg\min f$:

**Proposition E.3.** *Let $X_t$ be the sequence of iterates generated by* (OMD) *with assumptions as in Theorem 3. Then, with probability* 1*, there exists a* (*possibly random*) *subsequence of $X_t$ which converges to* $\arg\min f$.

*Proof.* Assume to the contrary that, with positive probability, the sequence $X_t$ generated by (OMD) admits no limit points in $\arg\min f$. Conditioning on this event, there exists a (nonempty) closed set $\mathcal{C} \subset \mathcal{X}$ which is separated by neighborhoods from $\arg\min f$ and is such that $X_t \in \mathcal{C}$ for all suffiently large $t$. Then, by relabeling $X_t$ if necessary, we can assume without loss of generality that $X_t \in \mathcal{C}$ for all $t \in \mathbb{N}$. Thus, by Proposition B.1, we get:

$$\Phi(x^*, Y_{t+1}) \leq \Phi(x^*, Y_t) - \gamma_t \langle v_t | X_t - x^* \rangle + \frac{\gamma_t^2}{2K} \|v_t\|_{X_t,*}^2$$

$$= \Phi(x^*, Y_t) - \gamma_t \langle \nabla f(X_t) | X_t - x^* \rangle - \gamma_t \langle U_{t+1} | X_t - x^* \rangle + \frac{\gamma_t^2}{2K} \|v_t\|_{X_t,*}^2$$

$$\leq \Phi(x^*, Y_t) - \gamma_t \delta(\mathcal{C}) + \gamma_t \xi_{t+1} + \frac{\gamma_t^2}{2K} \|v_t\|_{X_t,*}^2 \tag{E.7}$$

where in the last line we set $\delta(\mathcal{C}) = \inf\{\langle \nabla f(x) | x - x^* \rangle : x^* \in \arg\min f, x \in \mathcal{C}\} > 0$ (by (SI)), $U_{t+1} = v_t - \nabla f(X_t)$, $\xi_{t+1} = -\langle U_{t+1} | X_t - x^* \rangle$ and $\beta_t = \sum_{i=1}^{t} \gamma_i$. Thus, by telescoping and factorizing we get:

$$\Phi(x^*, Y_{t+1}) \leq \Phi(x^*, Y_1) - \beta_t \left[ \delta(\mathcal{C}) - \frac{\sum_{s=1}^{t} \gamma_s \xi_{s+1}}{\beta_t} - \frac{\sum_{s=1}^{t} \gamma_s^2 \|v_s\|_{X_s,*}^2}{2K\beta_t} \right] \tag{E.8}$$

By the unbiasedness assumption for $U_t$, we have $\mathbb{E}[\xi_{t+1} \mid \mathcal{F}_t] = \langle \mathbb{E}[U_{t+1} \mid \mathcal{F}_t] | X_t - x^* \rangle = 0$. Moreover, for all $x^* \in \arg\min f$, we have

$$\sum_{t=1}^{\infty} \gamma_t^2 \, \mathbb{E}[\xi_{t+1} \mid \mathcal{F}_t] \le \sum_{t=1}^{\infty} \gamma_t^2 \|X_t - x^*\|_{X_t}^2 \, \mathbb{E}[U_{t+1} \mid \mathcal{F}_t] \le \sum_{t=1}^{\infty} \gamma_t^2 \Phi(x^*, Y_t) \, \mathbb{E}[U_{t+1} \mid \mathcal{F}_t] < \infty \qquad \text{(E.9)}$$

where the last (strict) inequality is obtained due to the finite mean square property, the boundness of $\Phi(x^*, Y_t)$ and the fact that $\sum_{t=1}^{\infty} \gamma_t^2 < \infty$. Thus, we can apply the law of large numbers for $L^2-$martingales stated above and conclude that $\beta_t^{-1} \sum_{s=1}^{t} \gamma_s \xi_{s+1}$ converges to 0 almost surely. On the other hand, for the term $S_{t+1} = \sum_{s=1}^{t} \gamma_s^2 \|v_s\|_{X_t,*}^2$, since $v_{s+1}$ is $\mathcal{F}_s$-measurable for all $s = 1, 2 \ldots, t-1$ we have:

$$\mathbb{E}[S_{t+1} \mid \mathcal{F}_t] = \mathbb{E}\left[ \sum_{i=1}^{t-1} \gamma_t^2 \|v_i\|_{x_i,*}^2 + \gamma_t^2 \|v_t\|_{X_t,*}^2 \;\middle|\; \mathcal{F}_t \right] = S_t + \gamma_t^2 \, \mathbb{E}\left[ \|v_t\|_{X_t,*}^2 \;\middle|\; \mathcal{F}_t \right] \ge S_t \qquad \text{(E.10)}$$

so $S_t$ is a submartingale with respect to $\mathcal{F}_t$. Furthermore, by the law of total expectation, we also get:

$$\mathbb{E}[S_{t+1}] = \mathbb{E}[\mathbb{E}[S_{t+1} \mid \mathcal{F}_t]] \le \sigma^2 \sum_{i=1}^{t} \gamma_i^2 \le \sigma^2 \sum_{t=1}^{\infty} \gamma_t^2 < \infty, \qquad \text{(E.11)}$$

implying that $S_t$ is bounded in $L^1$. Thus, due to Doob's submartingale convergence theorem [15], we coclude that $S_t$ converges to some (almost surely finite) random variable $S_\infty$ so $\lim_{t\to\infty} \frac{S_{t+1}}{\beta_t} = 0$ with probability 1.

Now, by letting $t \to \infty$ in (E.8), we get $\Phi(x^*, Y_t) \to -\infty$, a contradiction. Going back to our original assumption, this shows that there exists a subsequence of $X_t$ which converges to $\arg\min f$ with probability 1, as claimed. ∎

With all this at hand, we proceed to the proof of our last-iterate convergence result:

*Proof of Theorem 3.* By the boundedness (and hence compactness) of $\arg\min f$, Proposition E.3 implies that, with probability 1, there exists some $x^* \in \arg\min f$ such that $X_{t_k} \to x^*$ for some (possibly random) subsequence $X_{t_k}$ of $X_t$. By the Riemann–Legendre property of $h$, it follows that $\Phi(x^*, Y_{t_k}) = D(x^*, X_{t_k}) \to 0$ as $k \to \infty$, implying in turn that $\lim_{t\to\infty} D(x^*, X_t) = 0$ (by Proposition E.1). Since $D(x^*, X_t) \ge K\|X_t - x^*\|_{X_t}^2 \ge \mu\|X_t - x^*\|^2$, we conclude that $X_t \to x^*$, and our proof is complete. ∎

# F   Applications to Poisson inverse problems and numerical experiments

## F.1   Detailed statement of the problem

The class of Poisson inverse problems that we consider stem from linear systems of the form

$$u = Hx + z \qquad \text{(F.1)}$$

where

- $x \in \mathbb{R}_+^d$ is the object under study (a signal, image, ...).
- $u \in \mathbb{R}_{++}^m$ is the observed data (usually $m \ll d$).
- The *kernel matrix* $H \in \mathbb{R}_+^{m \times d}$ is a representation of the data-gathering protocol and is typically highly ill-conditioned (e.g., a Toeplitz matrix in the case of image deconvolution problems).
- $z \in \mathbb{R}^m$ is the noise affecting the measurements.

When data points are obtained by means of a counting process, measurements can be modeled as Poisson random variables of the form $u_j \sim \text{Pois}(Hx)_j$. Then, up to an additive constant, the log-likelihood of $x \in \mathbb{R}^d$ given an observation $u \in \mathbb{R}_{++}^m$ will be

$$L(x; u) = -\sum_{j=1}^{m} \left[ u_j \log \frac{u_j}{(Hx)_j} + (Hx)_j - u_j \right]. \qquad \text{(F.2)}$$

Hence, obtaining a maximum likelihood estimate for $x$ leads to the archetypal *Poisson inverse problem*

$$\text{minimize} \quad f(x) \equiv D_{\mathrm{KL}}(u, Hx),$$
$$\text{subject to} \quad x \in \mathbb{R}^d_+,$$

(PIP)

where $D_{\mathrm{KL}}(p, q) = \sum_{j=1}^{m}[p_j \log(p_j/q_j) + q_j - p_j]$ denotes the generalized KL divergence on $\mathbb{R}^m_+$. For an extensive review of Poisson inverse problems, we refer the reader to Bertero et al. [7].

In many cases of practical interest, measurements arrive in distinct batches over time – e.g., as sequential optical sections in microscopy and tomography. Moreover, due to the large numbers of pixels/voxels involved (a typical range of values for $m$ is between $10^6$ and $10^7$), gradients of $f$ are very costly to compute; as such, optimization methods that rely on accurate gradient data are difficult to apply in this setting. Accordingly, a natural workaround to this obstacle is to exploit the online nature of the measurement process, model (PIP) as an *online* optimization problem, and then to use an online-to-batch conversion to get a candidate solution [35].

On the downside, this online optimization analysis crucially requires the loss functions faced by the optimizer to be Lipschitz continuous. However, this assumption does not hold for (PIP): if $f_j(x) = -u_j \log(u_j/(Hx)_j)$ denotes the singular part of the KL divergence for the $j$-th sample, we readily get

$$\frac{\partial f_j}{\partial x_i} = \frac{u_j H_{ji}}{(Hx)_j}.$$

(F.3)

This shows that the gradient of $f_j$ exhibits an $\mathcal{O}(1/x)$ singularity at the boundary of $\mathbb{R}^d_+$, so $f$ cannot be Lipschitz under *any* global norm on $\mathbb{R}^d$.

As suggested by Example 3, this singularity can be lifted by considering the *local* norm

$$\|z\|^2_x = (x_1 + \cdots + x_d)^2 \sum_{i=1}^{d} z_i^2 \quad \text{for all } v \in \mathbb{R}^d.$$

(F.4)

In this case, we have

$$\|\nabla f_j(x)\|^2_x = \sum_{i=1}^{d} u_j^2 \frac{H_{ji}^2 x_i^2}{(Hx)_j^2} = \frac{u_j^2 \sum_{i=1}^{d} H_{ji}^2 x_i^2}{\left[\sum_{i=1}^{d} H_{ji} x_i\right]^2} = \mathcal{O}(u_j^2),$$

(F.5)

so $\|\nabla f_j\|_x$ *is* bounded under this modified norm. This is the principal motivation for considering the Riemannian mirror descent method defined with respect to this metric and the regularizer presented in Example 5.

## F.2 Details on the experiments

In the rest of this appendix, we discuss in more detail the algorithms tested in Section 6. The algorithms we considered are

1. The *accelerated Lucy–Richardson algorithm*, as presented in [7] and corresponding to OMD with the entropic regularizer $h(x) = \sum_i x_i \log x_i$.
2. The composite mirror prox (CMP) of He et al. [18], corresponding to OMD with an extra gradient step and the log-barrier (Burg) regularizer $h(x) = -\sum_i \log x_i$ of Example 4.
3. The Riemannian mirror descent (RMD) algorithm detailed in Section 6, corresponding to the Poincaré-like regularizer of Example 5.

All algorithms were run with stochastic gradients drawn with the same minibatch size ($n = 256$) and a step-size of the form $\gamma_t \propto 1/\sqrt{t}$ (corresponding to the stochastic variant of each algorithm). For comparison purposes, we harvested each algorithm's last generated sample ("last iterate") as well as the corresponding ergodic average (defined here as $\bar{X}_T = \sum_{t=1}^{T} \gamma_t X_t / \sum_{t=1}^{T} \gamma_t$). Overall, the algorithms' last generated sample provided consistently better results than the ergodic average. The ground truth and the evolution of the Poisson likelihood loss are all reported in Fig. 1.

*Remark.* We should note here that the method of He et al. [18] can be seen as an "extra-gradient" version of the NoLips algorithm of Bauschke et al. [3] and the "relative stochastic gradient descent" scheme of Hanzely and Richtárik [16] (the difference between the last two being the step-size policy). In our experiments, Burg mirror descent with and without an extra-gradient step behaved similarly, with the extra-gradient version (CMP) performing slightly better. To minimize clutter, and because we are already comparing RMD to CMP above, we do not report this extra set of numerical experiments.

