# OpenReview forum: "Online and stochastic optimization beyond Lipschitz continuity: A Riemannian approach"
_ICLR.cc/2020/Conference — Accept (Spotlight)_

### Official Review · AnonReviewer1 · 2019-10-21
**Official Blind Review #1**

**Rating:** 6

**Review:**

This paper investigates online and stochastic convex optimization problems in which the objective function is not Lipschitz continuous. The originality of this study lies in the use of Riemannian geometry. Specifically, the standard condition of Lipschitz continuity is replaced with a more general condition involving Riemannian distances and called Riemann-Lipschitz Continuity (RLC). Based on an appropriate definition of Riemannian regularizer and a generalization of Fenchel coupling to Riemannian geometry, the authors provide $O(\sqrt T)$ regret (resp. risk) bounds for the online (resp. stochastic) mirror descent algorithm, under the Riemann-Lipchitz condition. The performance of the algorithm is validated on Poisson inverse problems.

Overall, this is a dense, yet interesting, paper. I am not an expert in Riemannian geometry but, as far as I could check, the proofs look correct. Notably, the analysis of OMD is relatively standard, once we get a bound (Prop B.1) on the Fenchel coupling, using the Riemannian dual norm.

I have essentially one main comment. Clearly, the concept of “Riemann-Lipschitz continuity” is different from the notions of “relative continuity” and “relative smoothness” that have been recently proposed in the literature. But it is not clear that the Riemann-Lipschitz condition can tackle convex optimization tasks in which relative continuity and relative smoothness do not hold. In particular, Poisson (linear) inverse problems have already been handled under the relative smoothness condition, using Mirror Descent or Bregman proximal methods (Hanzely and Richtarik, 2018; Hanzely et. al. 2018). Thus,
* from a conceptual viewpoint, it would be interesting to provide some applications in which the RLC condition hold, but the relative smoothness condition does not;
* from an experimental viewpoint, in Sec. 6, it would be legitimate to compare the present “Riemannian Mirror Descent” algorithm with respect to the APBG method (Hanzely et. al. 2018) and the relSGD method (Hanzely and Richtarik, 2018).


**Experience Assessment:**

I have read many papers in this area.

**Review Assessment: Checking Correctness Of Derivations And Theory:**

I assessed the sensibility of the derivations and theory.

**Review Assessment: Checking Correctness Of Experiments:**

I assessed the sensibility of the experiments.

**Review Assessment: Thoroughness In Paper Reading:**

I read the paper at least twice and used my best judgement in assessing the paper.

---

> ### Author Response · Authors · 2019-11-10
> **Thanks for the constructive feedback and positive evaluation!**
>
> Thank you for your constructive feedback and positive evaluation! Regarding your comments:
>
> 1.  We fully agree that the connection between non-Euclidean smoothness/continuity conditions is not always clear. For instance, depending on the choice of regularizer, a convex function which is differentiable on an open domain of $\mathbb{R}^n$ could be RL continuous, relatively continuous / smooth, or any combination of the above (or, of course, none).  In our view, this shows that a judicious choice of regularizer can lead to significant algorithmic performance gains (as different settings have different advantages/disadvantages). Our primary motivation for introducing the RLC condition (and hence extending the standard bounded gradient regularity assumption) was to focus on online and/or stochastic convex optimization problems where smoothness does not contribute to better regret rates. Going forward, we believe that a precise characterization of the interplay between the various smoothness/continuity conditions above would be of clear and certain value to the community - but, at the same time, this cannot be attempted within the scope of the current paper. We've introduced a "concluding remarks" section to discuss precisely this issue.
>
>
> 2.  Concerning the papers mentioned: we were not aware of the relSGD paper of Hanzely and Richtarik (2018), many thanks for bringing it to our attention! We have now included this paper in our review of the state of theart in the introduction - thanks again.
>
>
>
> 3.  Regarding the experimental part: since acceleration cannot be achieved in a generic stochastic setting without some variance reduction scheme (such as SVRG or the like), it is not clear how to put the APBG framework of Hanzely et al. on an equal footing with the methods studied in the current paper, so we did not attempt it. On the other hand, there are indeed several interesting connections with the recent paper of Hanzely and Richtarik (2018), which we detail below:
>
> 3a) First, we noticed a typo in p. 18 (now p.19 in Sec. F.2) of the supplement of our paper: when we referred to the Burg regularizer, we actually gave the definition of the standard (Gibbs-Shannon) entropic regularizer. That was a mistake, apologies for any confusion caused: the definition should read $h(x) = - \sum_{i} \log x_{i}$.
>
> 3b) To connect this with the work of Hanzely and Richtarik (2018), note that relSGD is  the mirror descent method generated by the Burg regularizer above with step-size $\gamma_t \propto 1/\sqrt{t}$.
>
> 3c) On that account, relSGD is most closely connected with the CMP method of He et al. (2016): CMP is also generated by the Burg entropy, but includes an extra-gradient step. [In Bregman language, relSGD is stochastic *mirror descent* with Burg regularization while CMP is stochastic *mirror-prox* with Burg regularization]
>
> 3d) CMP was one of the algorithms that we tested, and it was the second-best to RMD (which is generated by the O(1/x) regularizer of Example 4).
>
> 3e) Even though we did not report these results, Burg mirror descent with and without an extra-gradient step (i.e., CMP and relSGD respectively) behave similarly, with the extra-gradient version (CMP) performing slightly better.
>
> Since we are already comparing RMD to CMP (and CMP and relSGD behave similarly), we feel that adding an extra set of experiments would only occlude the discussion. Because of this, we opted not to present more experiments in the revised version of our manuscript; however, we are including a detailed version of the above discussion in the experimental section (p. 19 in the appendix), and we also discuss relSGD as an example of OMD in Section 4.
>
> For your convenience, we've outlined all relevant changes in our revision in blue.

---

### Official Review · AnonReviewer2 · 2019-10-27
**Official Blind Review #2**

**Rating:** 8

**Review:**

Summary:

The paper generalizes regret analysis results from convex online learning to
functions that are not Lipschitz but Riemann Lipschitz continuous.

Example:
$f(x) = -\log(x) + x$ is convex on the convex domain $X = [0, 2]$ but $f$ is
not Lipschitz on $D$ since $f(x) \to \infty$ as $x \to 0$.

A possible Riemannian metric that can be used on the domain of such a function
is the Poincare metric $g(x) = 1 / x^2$.
The norm defined by that Riemannian metric is $\| z \|^2_x = z^T g(x) z$.
This norm can be used to measure infinitesimal distances in $X$.
The Riemannian distance $dist(x, x')$ on $X$ that follows from this is defined
as integrating over the infinitesimal distances on a curve connecting
$x$ and $x'$ as measured by the norm defined above.

Intuitively, an infinitesimal distance Lipschitz bound can be seen to be
constructed by
$| f(x + \delta x) - f(x) | \approx \| f'(x) \| \| \delta x \|$ as $\delta x \to 0$
which in the case of our example does not exist since $f'(x) \to \infty$.
But when using a Riemannian metric based norm
$\| f'(x) \| = \sqrt{ f'(x)^T g^{-1}(x) f'(x) }$
on the right-hand side we have
$f'(x) = -\frac{1}{x} + 1$
$f'(x)^2 = \frac{1}{x^2} - \frac{2}{x} + 1$
$f'(x)^2 g^{-1}(x) = O(1)$
with $g(x) = 1 / x^2$.

In this way it is possible to bound changes of $f(x)$ relative to changes in
$x$ for functions that are not Lipschitz continuous.

The authors show how a suitable Riemannian metric can be transformed into
a regularizer usuable in online optimization.
They present various rates that appear to be otherwise known for similar
regularizers.
My knowledge of the online learning literature is very limited so I cannot
make a qualified statement about these formal analyses in a reasonable amount
of time.

The authors also transfer the results from the convex online setting to the
convex stochastic setting and the nonconvex setting.

Based on my limited understanding I would recommed to accept the paper.
The analysis to me seems both rigorous and useful in practice
(at least with regard to the formal definitions of Riemannian metrics and
Riemannian Lipschitz condition for singular functions).

Remarks / Suggestions:
- With a similar knowledge of the underlying function it would perhaps be
	possible to perform a nonlinear transformation of the input space that
	leads to Lipschitz continuous function.
	Can something be said about this?

- Definition 2: Write out l.s.c. (lower semi-continuous?) as it does not seem
	to be defined everywhere and not every reader is necessarily familiar
	enough with convex analysis

- Page 8: Typo: "Eucldiean stochastic gradient method


**Experience Assessment:**

I do not know much about this area.

**Review Assessment: Checking Correctness Of Derivations And Theory:**

I assessed the sensibility of the derivations and theory.

**Review Assessment: Checking Correctness Of Experiments:**

I did not assess the experiments.

**Review Assessment: Thoroughness In Paper Reading:**

I made a quick assessment of this paper.

---

> ### Author Response · Authors · 2019-11-10
> **Thanks for the constructive feedback and positive evaluation!**
>
> Many thanks for the constructive feedback and positive recommendation! Concerning your questions and remarks:
>
> 1. Indeed, a suitable nonlinear transformation of the feasible region could allow us to recover Lipschitz continuity in the standard (global) sense. In general however, a non-linear transformation would also destroy convexity, so we see no systematic way of transforming the problem in a way that would allow us to maintain both convexity and Lipschitz continuity.
>
> 2. Point taken about the definition of lower semicontinuity. In the revised version of our paper, we provide a clear definition at the point where the notion is introduced (bottom of p. 5).
>
> 3. Typos: fixed, many thanks!
>
> For your convenience, we've outlined all changes in our manuscript in blue.

---

### Official Review · AnonReviewer4 · 2019-11-02
**Official Blind Review #4**

**Rating:** 8

**Review:**

The paper establishes optimal regret bounds of the order O(\sqrt{T}) for Follow The Regularised Leader (FTRL) and Online Mirror Descent (OMD) for convex loss functions and potentials (a.k.a. Riemannian regularizers) that are, respectively, Lipschitz continuous and strongly convex with respect to a given Riemannian metric. These conditions naturally generalize the classical conditions typically considered in the literature, which are defined with respect to a global norm and, as such, are not well-suited to problems where the loss functions and its gradient present singularities at the boundary of the feasibility region. The authors suggest a principled way to choose both the Riemannian metric and the potential function based on the singularity landscape of the gradient of the loss function. Via standard online-to-batch conversion, the authors also address the offline setting and give O(1/\sqrt{T}) error bounds for ergodic averages in convex problems and for last iterates in non-convex problems satisfying a weak secant inequality. The authors include numerical experiments involving a Poisson inverse problem.

The paper is well-written, with a very clean narrative highlighting the main ideas and results. To the best of my knowledge, the literature review is complete and rightly highlights the fact that most results in the literature on Riemannian mirror descent methods have so far primarily addressed offline deterministic problems with exact oracle gradients. The contribution of this work lies not only in the focus on online and noisy setting but also on establishing natural results upon natural generalizations of well-known conditions in standard (non-Riemannian) settings. The techniques used are extensions of the classical theory and follow quite naturally, with the exception of the non-trivial primal-dual inequality (20).

QUESTIONS/SUGGESTIONS:
1) Can the authors be more explicit about how the (OMD) equations are derived from (16). While this is standard, I feel currently there is a bit of a jump in the narrative—which otherwise is very good.
2) The workhorse behind the established results is the primal-dual inequality (20) which relies on the introduction of the Fenchel coupling. Can the authors be more explicit about the use of this inequality, and what makes the Riemann generalization difficult in general? In particular, can the authors comment on the applicability of this inequality (or similar) to the smooth setting?
3) Typo: sometimes the notation $\mathcal{U}$ seems to be used instead of the notation $\mathcal{X}$. See, for instance, equation (7) in Definition 1 and the definition of the set $\mathcal{Z}$ in Remark 2.

AFTER REBUTTAL: I thank the authors for addressing my questions.

**Experience Assessment:**

I have read many papers in this area.

**Review Assessment: Checking Correctness Of Derivations And Theory:**

I assessed the sensibility of the derivations and theory.

**Review Assessment: Checking Correctness Of Experiments:**

I assessed the sensibility of the experiments.

**Review Assessment: Thoroughness In Paper Reading:**

I read the paper thoroughly.

---

> ### Author Response · Authors · 2019-11-10
> **Thanks for the thoughtful remarks and positive evaluation!**
>
> Many thanks for the constructive feedback and positive recommendation! Regarding your  questions and suggestions:
>
> 1. We will be happy to provide more details about the derivation of (OMD) from the “linearized” version of (FTRL). In the updated version of the manuscript, we have included a relevant paragraph right after (16).
>
> 2. The reviewer is correct that the primary workhorse of our analysis is the primal-dual inequality (20) for the Fenchel coupling. The main idea behind it serves a dual purpose: first, it seeks to leverage the generalized strong convexity condition in a way that allows the associated regularizer h to be properly adjusted to the singularity landscape of the problem. Our first idea here (which seemed easier to be checked), would be to assume that the Riemannian position-dependent norm would be applied at the base point (instead of x). However, the analysis breaks down completely under this modification. We have not been able to find another distance-like measure that is well-suited to the analysis of mirror descent in this context. The second point is that we need to compare primal vectors (solution points) to dual vectors (the iterates $Y_t$ that generate the algorithm): primal-primal divergence measures cannot do that, but the Fenchel coupling is perfectly suited for that purpose.
>
> The above also partially answers the reviewer’s comment regarding smoothness. In particular, a natural way to define smoothness via the use of a local Riemannian norm would be to extend the standard (Euclidean) descent lemma, namely by asking that $f(y) \leq f(x) + \langle\nabla f(x),y-x\rangle + L\|x-y\|_{y}^2$ for all $x,y$. However, if one applies a Riemannian gradient descent algorithm of the form $X_{t+1} = X_{t} - \gamma_{t} \mathrm{grad} f(X_{t})$ (with $\mathrm{grad} f$ denoting the standard Riemannian gradient),  the above definition would not offer a way to control the error quantity $-\gamma_{t}\|\mathrm{grad} f(X_t)\|_{X_t}^2 + L\gamma^{2}_{t}\|\mathrm{grad} f(X_{t})\|_{t+1}^2$, which is the main ingredient for obtaining the $O(1/T)$ convergence rate in the standard Euclidean case. This is circumvented by the use of the Lipschitz-like / relative smoothness discussed in the papers mentioned in the introduction (and also mentioned by Reviewer 1); however, as can be seen in these works, smoothness does not seem to provide any benefit in the online/stochastic case (where the best achievable rates are $O(1/\sqrt{T})$).
>
> 3. Typos: fixed, many thanks!
>
> For your convenience, we have highlighted all the relevant changes in our paper in blue.

---

### Decision · Program_Chairs · 2019-12-19

**Decision:**

Accept (Spotlight)

**Comment:**

This is a mostly theoretical paper concerning online and stochastic optimization for convex loss functions that are not Lipschitz continuous. The authors propose a method for replacing the Lipschitz continuity condition with a more general Riemann-Lipschitz continuity condition, under which they are able to provide regret bounds for the online mirror descent algorithm, as well as extending to the stochastic setting. They follow up by evaluating their algorithm on Poisson inverse problems.

The reviewers all agree that this is a well-written paper that makes a clear contribution. To the best of our knowledge, the theory and derivations are correct, and the authors were highly responsive to reviewers’ (minor) comments. I’m therefore happy to recommend acceptance.